# LabelFormer: Object Trajectory Refinement for Offboard Perception from LiDAR Point Clouds

**Anqi Joyce Yang** [1,2]   **Sergio Casas** [1,2]   **Nikita Dvornik**[1]   **Sean Segal**[1,2]
**Yuwen Xiong**[2*]   **Jordan Sir Kwang Hu**[1]   **Carter Fang**[1]   **Raquel Urtasun**[1,2]

Waabi[1]   University of Toronto[2]
{jyang, sergio, ndvornik, ssegal, jkhu, cfang, urtasun}@waabi.ai

**Abstract:** A major bottleneck to scaling-up training of self-driving perception systems are the human annotations required for supervision. A promising alternative is to leverage "auto-labelling" offboard perception models that are trained to automatically generate annotations from raw LiDAR point clouds at a fraction of the cost. Auto-labels are most commonly generated via a two-stage approach – first objects are detected and tracked over time, and then each object trajectory is passed to a learned refinement model to improve accuracy. Since existing refinement models are overly complex and lack advanced temporal reasoning capabilities, in this work we propose *LabelFormer*, a simple, efficient, and effective trajectory-level refinement approach. Our approach first encodes each frame's observations separately, then exploits self-attention to reason about the trajectory with full temporal context, and finally decodes the refined object size and per-frame poses. Evaluation on both urban and highway datasets demonstrates that *LabelFormer* outperforms existing works by a large margin. Finally, we show that training on a dataset augmented with auto-labels generated by our method leads to improved downstream detection performance compared to existing methods. Please visit the project website for details https://waabi.ai/labelformer/.

**Keywords:** Auto-label, Offboard Perception, Trajectory Refinement, Transformer

## 1   Introduction

Modern self-driving systems often rely on large-scale manually annotated datasets to train object detectors to perceive the traffic participants in the scene. Recently, there has been a growing interest in auto-labelling approaches that can automatically generate labels from sensor data. If the computing cost of auto-labelling is lower than the cost of human annotation and the produced labels are of similar quality, then auto-labelling can be used to generate much larger datasets at a fraction of the cost. These auto-labelled datasets can in turn be used to train more accurate perception models.

Following [1, 2], we use LiDAR as input as it is the primary sensor deployed on many self-driving platforms [3, 4]. In addition, we focus on the supervised setting where a set of ground-truth labels are available to train the auto-labeller. This problem setting is also referred to as offboard perception [2], which, unlike onboard perception, has access to future observations and does not have real-time constraints. Inspired by the human annotation workflow, the most common paradigm [1, 2] tackles the offboard perception problem in two stages, as shown in Fig. 1. First, objects and their coarse bounding box trajectories are obtained using a "detect-then-track" framework, and then each object track is refined independently. The main goal of the first stage is to track as many objects in the scene as possible (*i.e.* to achieve high recall), while the second stage focuses on track refinement to produce bounding boxes of higher quality. In this paper, we focus on the second stage, which we refer to as *trajectory refinement*. This task is challenging as it requires handling object occlusions, the sparsity of observations as range increases, and the diverse size and motion profiles of objects.

In order to handle these challenges, it is key to design a model that is able to effectively and efficiently exploit the temporal context of the entire object trajectory. However, existing methods [1, 2]

---

*Work done at Waabi.

7th Conference on Robot Learning (CoRL 2023), Atlanta, USA.

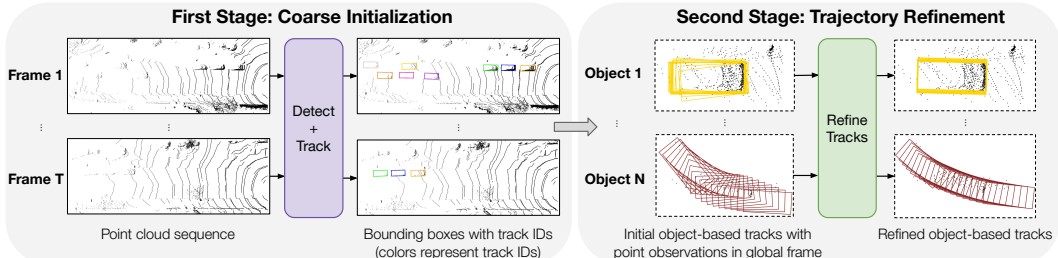

Figure 1: **Two-stage auto-labelling paradigm**. The first stage uses a detect-then-track paradigm to obtain coarse object trajectories. The second stage refines each trajectory independently.

fall short as they are designed to process trajectories from dynamic objects in a sub-optimal sliding window fashion, where a neural network is applied independently at each time step over a limited temporal context to extract features. This is inefficient as features from the same frame are extracted multiple times for different overlapping windows. As a consequence, the architectures exploit very small temporal context to remain within the computational budget. Furthermore, prior works utilized complicated pipelines with multiple separate networks (*e.g.*, to handle static and dynamic objects differently), which are hard to implement, debug, and maintain.

In this paper, we take a different approach and propose *LabelFormer*, a simple, efficient, and effective trajectory refinement method. It leverages the full temporal context and results in more accurate bounding boxes. Moreover, our approach is more computationally efficient than the existing window-based methods, giving auto-labelling a clear advantage over human annotation. To achieve this goal, we design a transformer-based architecture [5], where we first encode the initial bounding box parameters and the LiDAR observations at each time step independently, and then utilize self-attention blocks to exploit dependencies across time. Since our method refines the entire trajectory in a single-shot fashion, it only needs to be applied once for each object track during inference without redundant computation. Additionally, our architecture naturally handles both static and dynamic objects, and it is much simpler than other approaches.

Our thorough experimental evaluation on both urban and highway datasets show that our approach not only achieves better performance than window-based methods but also is much faster. Moreover, we show that LabelFormer can be used to auto-label a larger dataset for training downstream object detectors, which results in more accurate detections comparing to training on human data alone or leveraging other auto-labellers.

## 2 Related Works

**LiDAR-based Auto-Labelling:** These approaches have emerged from the need to automate the expensive human labelling process. Multiple approaches [6, 7, 8, 9, 10, 11, 12, 13] have attempted to generate auto-labels with little or no supervision, but they do not achieve satisfactory performance at high precision. More related to our method, pioneering works Auto4D [1] and 3DAL [2] developed a two-stage paradigm which first uses an object detector followed by a multi-object tracker to generate coarse object trajectories, and then refines each object trajectory separately using a supervised model. For the second-stage trajectory refinement, which is the problem we focus on in this work, Auto4D [1] consists of two separate models. It first trains a size branch that refines only the size with full temporal context, and then freezes the refined size and trains a motion path network to refine each frame's pose with a small local window. 3DAL [2] first trains a motion classifier, and then employs two separate networks for stationary and dynamic objects. The stationary network uses observations from the full trajectory, while the dynamic network again operates in a windowed fashion that consumes point cloud observations from a very small local window. As a result, both works have limited temporal context, incur heavier computational costs from overlapping windows, and involve complicated workflows with at least two models that cannot be trained jointly. In contrast, our work designs a single network to jointly optimize the entire bounding box trajectory at once. Finally, concurrent works [14, 15] also ingest the full object trajectory for refinement, but [14]

aggregates all point observations in the global frame and does not conduct explicit cross-frame temporal reasoning, and [15] trains three separate models while we only need one.

**Sequence processing with Transformers:** The transformer architecture [5] is the state-of-the-art model for sequence processing. It incorporates a self-attention mechanism that models communication between the sequence elements and enables sequence-level reasoning. Transformers have been successfully applied in various domains, including language modeling [5, 16, 17], image and video processing [18, 19], object tracking [20], and 3D understanding [21, 22]. They are also effective for multi-modal problems like vision-language joint modeling [23] and autonomous driving [24]. The application of transformers is particularly advantageous for modeling sequences with long-range dependencies, which is precisely the case for our trajectory-level refinement task. To handle long sequences, it is common to preprocess the input tokens with a powerful short-range encoder and then utilize a transformer to perform global reasoning on the encoder's outputs [25, 26, 27]. For instance, the work of [25] addresses long video modeling by processing short video snippets with object detection, tracking, and action recognition models before passing their outputs to a transformer for holistic video analysis. Drawing inspiration from these works, our LabelFormer takes the per-frame bounding box initializations obtained with a detector and a tracker, and refines them by jointly considering the bounding box trajectories and point clouds of the entire trajectory sequence.

## 3 LabelFormer: Transformer-based Trajectory-level Refinement

The goal of trajectory refinement is to produce an accurate bounding box trajectory given a noisy initialization that is typically obtained using a detect-then-track paradigm. In this paper we propose a novel transformer-based architecture that takes the raw sensor observations and the full object trajectory as input, and conducts temporal reasoning simultaneously for all frames in the trajectory. This architecture naturally handles both static and dynamic objects and jointly refines the bounding box size and the motion path, resulting in a much simpler, efficient and effective approach.

### 3.1 Problem Setting

The input to a LiDAR-based auto-labeller is a sequence of $T$ "frames" of point clouds, obtained from a single LiDAR scan (*i.e.*, a 360° sweep). Since Bird's-Eye View (BEV) is the de-facto representation for downstream tasks in self-driving, such as motion and occupancy forecasting [28, 29, 30, 31, 32] and motion planning [33, 34, 35, 36], LabelFormer operates in BEV.

In the first stage of the auto-labelling pipeline, we use a detection model followed by a multi-object tracker, which provide us with $N$ perceived objects with initial bounding box trajectories $(\mathbf{B}^1, \ldots, \mathbf{B}^N)$. Each object trajectory $\mathbf{B}$ (we omit the object index for brevity) is defined by a sequence of $M$ bounding boxes $\mathbf{B} = (\mathbf{b}_1, \ldots, \mathbf{b}_M)$, where each BEV bounding box $\mathbf{b}_i = (x_i, y_i, l_i, w_i, \theta_i)$ is parameterized by center position $(x_i, y_i)$, bounding box length and width $(l_i, w_i)$, and heading angle $\theta_i$. All bounding box poses $(x_i, y_i, \theta_i)$ are in a *trajectory coordinate frame* that is centered at the middle of the trajectory, *i.e.*, $(x_m, y_m, \theta_m) = (0, 0, 0)$ with $m = M//2$ as the middle index. Note that the sequence length $M$ may vary across objects, and that the bounding box dimensions $(l_i, w_i)$ for the same object might be different across frames because object detectors output bounding boxes for each frame separately. In addition, each bounding box $\mathbf{b}_i$ is detected from a scene-level point cloud $\mathbf{P}_i \in \mathbb{R}^{n_i \times 4}$ that consists of $n_i$ points, and each point is represented as its 3D position in the same trajectory frame along with its timestamp.

Given each object's coarse bounding box trajectory $\mathbf{B} = (\mathbf{b}_1, \ldots, \mathbf{b}_M)$ and the scene-level point clouds $(\mathbf{P}_1, \ldots, \mathbf{P}_M)$, the goal of trajectory refinement is then to output a precise trajectory $\hat{\mathbf{B}} = (\hat{\mathbf{b}}_1, \ldots, \hat{\mathbf{b}}_M)$ with a bounding box size $(\hat{l}, \hat{w})$ that is shared across the entire trajectory.

### 3.2 Model Architecture

To refine the entire actor trajectory with full temporal context, *LabelFormer* first uses a shared encoder to process each frame's observations separately, and then leverages self-attention to reason across frames. Finally, a decoder is employed to obtain the refined pose at each frame as well as a consistent bounding box size for the full duration of the trajectory. Fig. 2 illustrates the architecture.

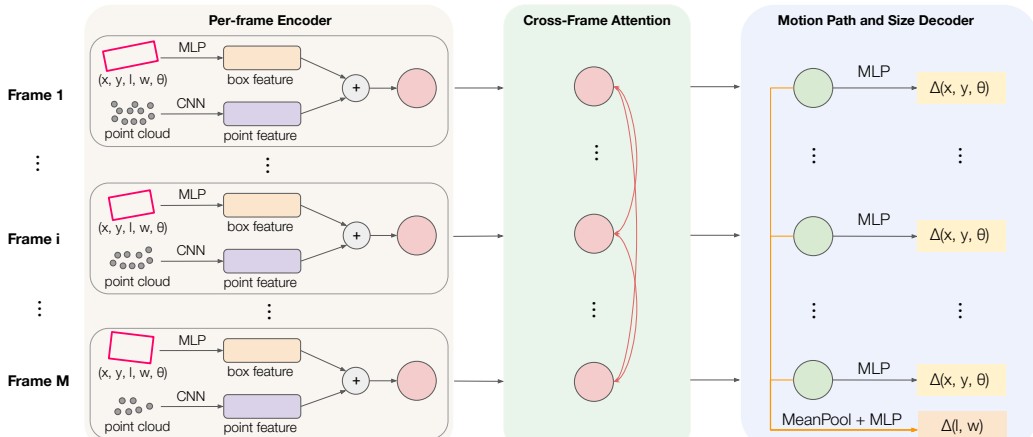

Figure 2: **LabelFormer Architecture** which first encodes box and point observations for each frame separately, then applies a stack of self-attention layers among per-frame features, and finally decode a size residual along with per-frame pose residuals.

### 3.2.1 Per-Frame Encoder

Given the initial object BEV bounding box trajectory $\mathbf{B} = (\mathbf{b}_1, \ldots, \mathbf{b}_M)$ and point clouds $(\mathbf{P}_1, \ldots, \mathbf{P}_M)$ in the trajectory frame described in the problem setting, we first extract object points inside each bounding box $\mathbf{b}_i$ by filtering the respective scene-level point cloud $\mathbf{P}_i$ in BEV. That is, we only keep the points in $\mathbf{P}_i$ whose BEV projection lies inside the 2D BEV bounding box $\mathbf{b}_i$. Since the bounding box initialization $\mathbf{b}_i$ is noisy, similar to previous works [1, 2], we enlarge the filtering region by $10\%$ such that the point cloud contains the full object with high probability. As a result, each frame $i$ has two input observations: the initial BEV bounding box $\mathbf{b}_i \in \mathbb{R}^5$ and the set of $m_i$ 3D object points $\mathbf{O}_i \in \mathbb{R}^{m_i \times 4}$. We next encode each frame's bounding box and point observations separately before fusing them. For brevity, we refer to "bounding box" as "box" from now on.

**Bounding Box Encoding:** Since the detector might produce noisy heading directions, we first pre-process the heading directions with a simple heuristic that flips inconsistent headings by $180°$ based on majority voting. We next use a simple multi-layer perceptron (MLP) that maps box parameters $\mathbf{b}_i$ with updated heading directions to high-dimensional features $\mathbf{a}_i = \mathrm{MLP}(\mathbf{b}_i) \in \mathbb{R}^D$.

**Point Cloud Encoding:** Previous works [1, 2] aggregate multi-frame points in the global frame for feature extraction. However, in practice, humans estimate the relative transformation between two point clouds by aligning them in the object frame. Motivated by this fact, we use the object pose initialization $\mathbf{b}_i$ to transform $\mathbf{O}_i$ from the trajectory frame to the object frame. We next learn a representation of the object-frame points with a PointPillars [37]-style encoder. Specifically, we first voxelize the object-frame points into a $N_x \times N_y \times N_z$ grid, apply a PointNet [38] to all points in each 3D voxel grid to extract per-voxel feature, and fuse all features along height to generate a BEV feature map $\in \mathbb{R}^{N_x \times N_y \times D_v}$. We next feed the BEV voxel feature map into a 2D convolutional network (CNN) composed of a multi-scale ResNet [39] backbone followed by FPN [40] to obtain a $4\times$ downsampled feature map $\mathbf{F}_i \in \mathbb{R}^{N'_x \times N'_y \times D'}$. Since the receptive field of the CNN is designed to cover the entire object space, to retrieve point feature $\mathbf{p}_i \in \mathbb{R}^{D'}$ we simply index the feature map $\mathbf{F}_i$ at its spatial center.

**Feature Fusion:** Finally, we fuse the box feature $\mathbf{a}_i$ and point feature $\mathbf{p}_i$ to derive the final frame-wise feature $\mathbf{f}_i = \mathbf{a}_i + \mathrm{MLP}(\mathbf{p}_i) \in \mathbb{R}^D$. We apply the same encoder with shared weights on each frame individually and obtain a set of frame-wise feature tokens $\{\mathbf{f}_i\}_{1 \leq i \leq M}$.

### 3.2.2 Trajectory-level Understanding via Cross-Frame Attention

Intuitively, both box parameters and point observations are useful for trajectory refinement. However, traditional path smoothers and point cloud registration methods based on ICP [41, 42] fail to fuse the information from both sources. In addition, ICP reasons on the point level and fails

when the points are sparse or the point cloud pair has small overlaps. Structured optimization-based approaches [7, 8] that combine both methods operate in an online fashion with limited temporal context, and still suffer from ICP's failure modes. To address these limitations from traditional methods, we exploit the fused per-frame features $(\mathbf{f}_1, \ldots, \mathbf{f}_M)$, and model relationships between frames at the feature level via self-attention [5]. The attention module allows for efficient pairwise reasoning across frames and offers the flexibility to operate on sequences with arbitrary length.

At a high level, the input to the attention module is the feature sequence $(\mathbf{f}_1, \ldots, \mathbf{f}_M)$, and the output $(\mathbf{g}_1^{(L)}, \ldots, \mathbf{g}_M^{(L)})$ represents the updated per-frame features after absorbing information from the entire sequence. In particular, the attention module contains $L$ attention blocks, where the $j^{\text{th}}$ attention block consumes the previous block's output feature sequence $(\mathbf{g}_1^{(j-1)}, \ldots, \mathbf{g}_M^{(j-1)})$ and generates an updated feature sequence $(\mathbf{g}_1^{(j)}, \ldots, \mathbf{g}_M^{(j)})$, with the first attention block input $\mathbf{g}_i^{(0)} = \mathbf{f}_i$. Each attention block contains a self-attention layer followed by a feed-forward MLP. For each input feature vector $\mathbf{g}_i^{(j-1)} \in \mathbb{R}^D$, the pre-norm self-attention mechanism first applies LayerNorm [43] (LN) followed by three separate linear projections to derive query, key and value vectors $\mathbf{q}_i^{(j)}, \mathbf{k}_i^{(j)}, \mathbf{v}_i^{(j)} \in \mathbb{R}^D$ respectively. We stack the keys and values from all $M$ frames to form matrices $\mathbf{K}^{(j)}, \mathbf{V}^{(j)} \in \mathbb{R}^{M \times D}$. Then, for each query frame $i$, we compute the attention scores between frame $i$ and all frames by comparing query vector $\mathbf{q}_i^{(j)}$ and each key in $\mathbf{K}^{(j)}$:

$$\mathbf{a}_i^{(j)} = \text{softmax}\left(\frac{\mathbf{q}_i^{(j)}\mathbf{K}^{(j)^T}}{\sqrt{d_k}} + \mathbf{w}_i\right) \in \mathbb{R}^M, \tag{1}$$

where $d_k$ is a scaling factor. Note that in Eq. 1 we adopt AliBi [44], which is a form of relative positional encoding that leverages positional difference between query and key frames. AliBi directly adds weighted biases $\mathbf{w}_i \in \mathbb{R}^M$ (with $w_{ij} = -m \cdot |i - j|$, $m$ is a fixed constant) to the dot product attention score map and is shown to generalize better to longer sequences at test time.

With the attention scores, we can then obtain an aggregated feature with $\mathbf{h}_i^{(j)} = \mathbf{a}_i^{(j)}\mathbf{V}^{(j)} \in \mathbb{R}^D$.

We then apply the subsequent MLP layer to derive output features $\mathbf{g}_i^{(j)}$ of the $j^{\text{th}}$ attention block:

$$\begin{aligned} \mathbf{h}_i'^{(j)} &= \text{LN}(\mathbf{g}_i^{(j-1)}) + \mathbf{h}_i^{(j)} \\ \mathbf{g}_i^{(j)} &= \text{MLP}(\text{LN}(\mathbf{h}_i'^{(j)})) \in \mathbb{R}^D. \end{aligned} \tag{2}$$

In practice, we use *multi-head* self-attention to increase expressivity, which partitions the $D$-dimensional input feature vector into $H$ groups, employs a separate self-attention head for each feature group, and concatenates the output features from each attention head as the final feature. Please refer to [5] for more details on multi-head attention. After $L$ chained self-attention blocks, the attention module outputs a sequence of updated feature vectors at each frame $(\mathbf{g}_1^{(L)}, \ldots, \mathbf{g}_M^{(L)})$, which is used to decode the final bounding box trajectory.

### 3.2.3 Motion Path and Size Decoder

Given the feature sequence $(\mathbf{g}_1^{(L)}, \ldots, \mathbf{g}_M^{(L)})$, we decode the final motion path trajectory and object size which is consistent for the entire trajectory. To decode the refined bounding box pose at each frame, we simply feed the feature $\mathbf{g}_i^{(L)}$ into an MLP to obtain pose residual $(\Delta x_i, \Delta y_i, \Delta \theta_i)$ and sum them with the initialization $\mathbf{b}_i$ to obtain the final refined pose parameters $(\hat{x}_i, \hat{y}_i, \hat{\theta}_i) = (x_i + \Delta x_i, y_i + \Delta y_i, \theta_i + \Delta \theta_i)$. To decode the refined object size, we leverage context from all frames via mean pooling, and use an MLP to obtain a size residual $(\Delta l, \Delta w) = \text{MLP}(\text{mean}(\{\mathbf{g}_i^{(L)}\}))$. We compute the final refined object size as $(l, w) = (\text{mean}(\{l_i\}) + \Delta l, \text{mean}(\{w_i\}) + \Delta w)$. The final refined bounding box at each frame $i$ will be $\hat{\mathbf{b}}_i = (\hat{x}_i, \hat{y}_i, \hat{l}, \hat{w}, \hat{\theta}_i)$.

### 3.3 Training

We train the entire model (*i.e.*, encoder, attention module, decoder) end-to-end by minimizing a combination of a regression loss that directly compares the refined box parameters $\hat{\mathbf{b}}_i$ with ground-truth box parameters $\mathbf{b}_i^\star$, and an IoU-based loss that compares the axis-aligned bounding boxes:

$$L(\{\hat{\mathbf{b}}_i\}, \{\mathbf{b}_i^\star\}) = L_{reg}(\{\hat{\mathbf{b}}_i\}, \{\mathbf{b}_i^\star\}) + L_{IoU}(\{\hat{\mathbf{b}}_i\}, \{\mathbf{b}_i^\star\}). \tag{3}$$

Please see supp. for more details. We apply two forms of data augmentation during training: (1) we randomly sample a subsequence of the input actor trajectory, and (2) we independently perturb each initial bounding box by applying a translational offset uniformly sampled from $[-0.25, 0.25]$m for $x$ and $y$ each, a rotational offset uniformly sampled from $[-10, 10]$ degrees, and an offset uniformly sampled from $[\max(-0.2, -\frac{l_i}{2}), \min(0.2, \frac{l_i}{2})]$ and $[\max(-0.1, -\frac{w_i}{2}), \min(0.1, \frac{w_i}{2})]$ for the dimensions. Note that the offsets are sampled per frame and applied to each bounding box separately.

## 4    Experiments

In this section, we evaluate the effectiveness of our approach on two real-world datasets. First, we describe the experimental setting and metrics used for evaluation. Next, we show that our method outperforms previous works on the trajectory refinement task for multiple initializations. Furthermore, we demonstrate that the improved refinement translates to downstream detection performance when training with auto-labels, showcasing that our approach can be used to train better object detectors. Finally, we conduct thorough ablation studies to analyze the effect of our design choices.

**Datasets:**    We use two datasets to evaluate our method in both urban and highway domains, which include object trajectories with diverse motion profiles. For the urban setting, we use the *Argoverse 2 Sensor dataset* [3] (AV2) that was collected in six distinct US cities. AV2 contains 850 15-second long snippets and around 65.7k vehicle trajectories. The LiDAR data is fairly sparse as it is captured by two 32-beam LiDARs that spin at 10Hz in the same direction but are $180°$ apart in orientation, and we aggregate both LiDAR scans in the same sweep interval to form an input frame. We use the official train and validation splits with 700 and 150 snippets each. For the highway setting, we use an in-house *Highway dataset*, which contains 188 20-second long snippets collected from US highways with roughly 5.8k vehicle trajectories. The LiDAR data in this dataset is denser as it comes from a 128-beam LiDAR sensor that spins at 10Hz. We split the dataset into 150 training snippets and 38 validation snippets. For our experiments in both datasets, we focus on auto-labelling vehicles in the scene, with a detection region of interest of [-125, 200] meters longitudinally and [-50, 50] meters laterally with respect to the ego vehicle's traveling direction.

**Metrics:**    Following [1], for each object $k$, we compute the track-level IoU $S^k = \frac{1}{M_k} \sum_{i=1}^{M_k} \text{IoU}(\mathbf{B}^{\star k}_i, \hat{\mathbf{B}}^k_i)$, where $M_k$ is the trajectory length and $\mathbf{B}^{\star k}_i, \hat{\mathbf{B}}^k_i \in \mathbb{R}^5$ are the respective ground-truth and refined BEV bounding box at each frame $i$. To aggregate across all $N$ object trajectories, we report the mean IoU as $\frac{1}{N} \sum_{k=1}^{N} S^k$. To understand coverage, we additionally report average recall at various IoU thresholds, *i.e.*, RC@$\alpha = \frac{1}{N} \sum_{k=1}^{N} \mathbf{1}(S^k \geq \alpha)$, where $\mathbf{1}(x \geq \alpha) = \begin{cases} 1 & \text{if } x \geq \alpha \\ 0 & \text{otherwise} \end{cases}$ is the indicator function. In our results we choose IoU thresholds $\alpha = 0.5, 0.6, 0.7, 0.8$.

**Implementation details:**    For the box encoder, we use a single linear layer to map the 5 box parameters to an output dimension, $D = 256$. For the attention module, we use $L = 6$ attention blocks, with $H = 4$ attention heads. Both the size and pose decoders are a single linear layer. For both datasets, we train our model with the AdamW optimizer [45], with learning rate 5e-5 and weight decay 1e-5. We apply linear learning rate warmup in the first two epochs followed by cosine learning rate scheduling that eventually decays the initial learning rate (after warmup) by $10\times$. We additionally clip the gradient norms at 5.0 as we empirically find this helps with learning. We train our model for 100 epochs on the Highway dataset and 40 epochs on the AV2 dataset with a batch size of 4. Please see supp. for more details on the point encoder and attention network.

**Baselines:**    We compare our proposed *LabelFormer* with state-of-the-art auto-labelling methods Auto4D [1] and 3DAL [2]. Since neither papers released their code, we reimplemented both methods based on their implementation details and thoroughly tuned the hyperparameters. We follow the original implementations to set the Auto4D's temporal window size to be 10. For 3DAL dynamic branch, we use the original bounding box temporal window size of 101 and raised the point window size from 5 to 11 to increase model performance. Note that since the Highway dataset contains very

| Dataset | First-Stage Detector | Second-Stage Refinement | Mean IoU | RC @ 0.5 | RC @ 0.6 | RC @ 0.7 | RC @ 0.8 |
|---|---|---|---|---|---|---|---|
| AV2 | PointPillars [37] | - | 62.60 | 76.39 | 65.72 | 48.66 | 25.04 |
| | | Auto4D [1] | 65.49 | 79.47 | 70.35 | 56.04 | 32.40 |
| | | 3DAL [2] | 64.58 | 77.25 | 67.92 | 53.93 | 32.53 |
| | | Ours (LabelFormer) | **68.28** | **81.22** | **73.22** | **60.68** | **40.78** |
| | VoxelNeXt [46] | - | 65.70 | 78.92 | 68.90 | 54.37 | 32.44 |
| | | Auto4D [1] | 67.92 | 81.73 | 73.14 | 59.42 | 37.02 |
| | | 3DAL [2] | 67.12 | 80.36 | 71.42 | 57.83 | 36.01 |
| | | Ours (LabelFormer) | **70.18** | **83.26** | **75.06** | **63.04** | **43.76** |
| Highway | PointPillars [37] | - | 60.20 | 69.83 | 59.62 | 46.11 | 26.58 |
| | | Auto4D [1] | 65.63 | 77.01 | 68.23 | 56.28 | 37.08 |
| | | 3DAL [2] | 66.03 | 79.44 | 70.33 | 56.62 | 35.18 |
| | | Ours (LabelFormer) | **70.59** | **83.09** | **75.77** | **65.11** | **46.16** |
| | VoxelNeXt [46] | - | 65.90 | 76.76 | 68.32 | 56.27 | 36.77 |
| | | Auto4D [1] | 69.27 | 79.36 | 72.81 | 63.37 | 45.39 |
| | | 3DAL [2] | 68.17 | 80.61 | 73.07 | 60.61 | 38.85 |
| | | Ours (LabelFormer) | **72.38** | **83.68** | **77.45** | **68.33** | **50.06** |

Table 1: **Comparison with state-of-the-art**

few static objects, we train the 3DAL dynamic branch with all objects in the scene and only apply the dynamic network during inference. For the AV2 dataset we apply the original 3DAL method with both stationary and dynamic branches.

**First-stage initialization:** We obtain the first-stage coarse initializations by running a detection and tracking model. For fair comparison between the refinement approaches, we train and evaluate all refinement models on the same set of true-positive first-stage object trajectories, *i.e.*, those that can be associated with a ground-truth object trajectory (more details in supp.). To ensure our conclusions generalize, for each dataset, we evaluate the refinement approaches on initializations from two detection models. Specifically, we experiment with a multi-frame version of PointPillars [37] and a recent state-of-the-art detector VoxelNeXt [46] as the first-stage detector. Following [1, 2], we implement a simple rule-based multi-object tracker, please see supp. for more details.

**Refinement results against SOTA:** Table 1 shows that in terms of refinement accuracy, our method consistently outperforms the initializations and state-of-the-art auto-label refinement methods by a large margin across both detector initializations on both datasets. While existing methods already have significant gains over the initializations, our method is able to achieve on

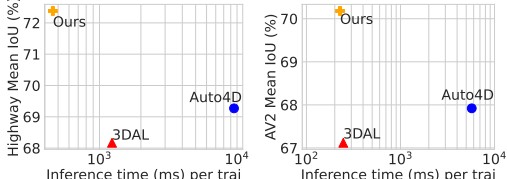

Figure 3: Refinement quality vs. runtime

average 92% more mean IoU gains. In addition, our method is able to achieve significantly higher accuracy on both static and dynamic objects with a single network, as shown in supp. Furthermore, we measure the average refinement run time on trajectories from the VoxelNeXt initialization, using a single RTX2080 GPU. Results are shown in Fig. 3 alongside mean IoU. We find that LabelFormer is $2.7\times$ faster than the window-based dynamic 3DAL on the Highway dataset. On AV2, which has 52% of static objects, our method is still slightly faster than 3DAL which uses a non-window-based static branch. Auto4D uses all available points while 3DAL samples at most 1024 points per frame, resulting in Auto4D having much longer run time but higher performance at large IoU thresholds.

**Improving object detection with auto-labeled data:** We additionally study the effect of the refined auto-labels in the downstream object detection task. Specifically, we use VoxelNeXt [46] as the first-stage detector and train various auto-labellers on the main Highway training set (consisted of 150 human-labelled

| Auto-Label | Mean AP | AP@0.5 | AP@0.7 | AP@0.8 |
|---|---|---|---|---|
| N/A | 82.98 | 91.62 | 79.17 | 55.97 |
| Init | 83.63 | 92.67 | 79.51 | 55.30 |
| Auto4D | 83.42 | 92.71 | 79.32 | 55.07 |
| 3DAL | 83.64 | 92.66 | 79.76 | 55.79 |
| Ours | **84.81** | **92.91** | **80.91** | **59.00** |

Table 2: **[Highway] Downstream task**

snippets), and use them to label an additional 500 Highway snippets. We then train a downstream object detector with a combined dataset of 150 human-labelled snippets and 500 auto-labelled snippets. Table 2 shows the average precision (AP) results. Overall, training with the bigger dataset augmented with auto-labels is better than with the human-labelled dataset alone, and our refined auto-labels give the biggest boost with a 3% gain at 0.8 IoU.

| | Box Enc. | Point Enc. | Perturb | Window | # Att | Mean IoU | RC@0.5 | RC@0.6 | RC@0.7 | RC@0.8 |
|---|---|---|---|---|---|---|---|---|---|---|
| $M_1$ | ✓ | | ✓ | All | 6 | 69.20 | 80.68 | 73.92 | 63.28 | 43.76 |
| $M_2$ | | ✓ | ✓ | All | 6 | 70.17 | 81.73 | 74.74 | 64.20 | 45.61 |
| $M_3$ | ✓ | ✓ | | All | 6 | 70.97 | 82.37 | 75.37 | 65.32 | 47.65 |
| $M_4$ | ✓ | ✓ | ✓ | All | 1 | 71.59 | 83.62 | 76.35 | 66.62 | 48.87 |
| $M_5$ | ✓ | ✓ | ✓ | All | 3 | 72.11 | **83.93** | **77.50** | 67.66 | 49.90 |
| $M_6$ | ✓ | ✓ | ✓ | 5 | 6 | 71.18 | 82.33 | 75.44 | 65.81 | 48.80 |
| $M_7$ | ✓ | ✓ | ✓ | 10 | 6 | 71.72 | 83.26 | 76.57 | 66.68 | 49.69 |
| $M_8$ | ✓ | ✓ | ✓ | 20 | 6 | 71.92 | 83.14 | 76.62 | 67.08 | 49.91 |
| $M_9$ | ✓ | ✓ | ✓ | All | 6 | **72.38** | 83.68 | 77.45 | **68.33** | **50.06** |

Table 3: **[Highway] Ablation study** using the VoxelNeXt initializations. Perturb refers to the bounding box perturbation augmentation, Window specifies the window size when applicable, and "# Att" is the number of self-attention blocks.

Initialization LabelFormer

Figure 4: **Qualitative results:** Initialization vs. refinement. Auto-labels in orange, GT in magenta.

**Effect of box and point features:** $M_1$ and $M_2$ in Table 3 each only encode box features and point features respectively. Comparing to $M_9$ which uses both features, we show that both the box and point features in the per-frame encoder stage contribute to the overall success of the model.

**Effect of per-frame perturbation:** $M_3 \to M_9$ in Table 3 shows that the per-frame bounding box perturbation augmentation we use helps with the final performance.

**Effect of number of self-attention blocks:** $M_4$, $M_5$ and $M_9$ in Table 3 show that the refinement accuracy grows with more attention blocks.

**Effect of temporal context length:** Finally, we train and run our method in a window-based fashion to restrict the temporal context given to each method. $M_6$, $M_7$, $M_8$ and $M_9$ in Table 3 show that the refinement accuracy steadily increases with more temporal context given to the model.

**Qualitative results:** Fig. 4 shows an example for which the observations are very sparse at the beginning of the trajectory and get denser afterwards. LabelFormer is able to exploit the full temporal context and improve the bounding box trajectory, especially at the most challenging frames. More examples and comparisons with previously proposed refinement methods can be found in the supp.

## 5 Limitations

The two-stage auto-labelling paradigm has an inherent limitation: the second stage only refines the continuous bounding box localization errors, but does not correct discrete detection errors (false positives, false negatives) and tracking errors (id switches, fragmented tracklets). Such discrete errors will propagate to the final output. Moreover, the refinement performance on the fragmented tracklets may be sub-optimal due to the missing temporal context (otherwise present in the unfragmented tracklets). Therefore, a future direction is to explore alternative paradigms that can recover from discrete errors too. Finally, a failure mode of our proposed refinement model is that it can degrade the quality of the auto-labels with respect to initialization when the input trajectories are short and have sparse observations. Such cases are challenging even to humans, yet future work can try to address this by estimating auto-label uncertainty and leveraging it in downstream applications.

## 6 Conclusion

In this work, we study the trajectory refinement problem in a two-stage LiDAR-based offboard perception paradigm. Our proposed method, *LabelFormer*, is a single transformer-based model that leverages full temporal context of the trajectory, fusing information from the initial path as well as the LiDAR observations. Compared to prior works, our method is simple, achieves much higher accuracy and runs faster in both urban and highway domains. With the ability to auto-label a larger dataset effectively and efficiently, LabelFormer helps boost downstream perception performance, and unleashes the possibility for better autonomy systems.

**Acknowledgments**

We thank the anonymous reviewers for the helpful comments and suggestions. We would also like to thank Bin Yang and Ioan Andrei Bârsan for insightful discussions and guidance at the early stage of the project. Finally, we would like to thank the Waabi team for their valuable support.

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

# Supplementary Materials

## A   Implementation and Experiment Details

### A.1   Model Details

**Point Encoder:**   The point encoder is consisted of a voxelizer followed by a CNN-based backbone and a Feature Pyramid Network (FPN) [40].

Specifically, the voxelizer employs voxel resolution of 10cm in all of $X$, $Y$ and $Z$ directions, with a region of interest of [-12, 12] meters along $X$, [-4, 4] meters along $Y$, and [-0.2, 3.0] meters along $Z$, to construct a $N_x \times N_y \times N_z$ voxel grid with $N_x = 240$, $N_y = 80$ and $N_z = 320$. For all points in each 3D voxel grid, we first represent each point as $(\Delta x, \Delta y, \Delta z, \Delta t)$ where $(\Delta x, \Delta y, \Delta z)$ is the positional offset with respect to the voxel centroid and $\Delta t = t - t_{ref}$ is the difference between the per-point time and the LiDAR sweep end time of the middle frame in the object trajectory. We feed this four-vector representation of each point into a two-layer MLP with 16 output channels each, and LayerNorm [43] and ReLU applied right after the first layer. Then, for each voxel, we pool all point features inside by summing them and applying a LayerNorm after to derive voxel grid features $\mathbb{R}^{N_x \times N_y \times N_z \times 16}$, which can be viewed as $N_x \times N_y$ "feature pillars" along the $Z$ axis. We additionally encode an $z$-axis positional embedding via a learnable variable block $\in \mathbb{R}^{N_z \times 16}$. We concatenate the non-empty voxels in each feature pillar with the positional embedding block to obtain an augmented feature $\in \mathbb{R}^{N_z' \times 32}$ ($N_z' \leq N_z$ is the number of non-empty voxels in the pillar), and pass through a two-layer MLP with 16 and 32 output channels each (with LayerNorm and ReLU in between), apply LayerNorm after the second layer, and sum all the features along each pillar to obtain a BEV feature map$\in \mathbb{R}^{N_x \times N_y \times 32}$.

The backbone takes the BEV feature map as input and first applied three stem layers with 120, 96, 96 output channels each. Each stem layer is consisted of a 3x3 convolutional layer, followed by GroupNorm (GN) and ReLU. The first stem layer has stride of 2 while the next two have a stride of 1. Then, the output passes through three downsampling stages. The three downsampling stages contain 6, 6, 4 ResNet [39] blocks with 288, 384, 576 output channels respectively. Each ResNet block applies a sequence of 1x1 conv, GN, ReLU, 3x3 conv (with an optional stride parameter), GN, ReLU, 1x1 conv, GN, ReLU to obtain a residual and sum with the input. Each downsampling stage downsamples the input by a factor of 2× within the first ResNet block, where the first ResNet block has stride 2 in the middle 3x3 conv, and the residual output is added to the input that is downsampled with a 1x1 conv block with stride 2 followed by GN. The remaining ResNet blocks in each downsampling stage all have stride of 1.

The FPN takes the outputs from all three stages in the backbone, which are 4×, 8× and 16× down-sampled from the original resolution (the stem layers downsample the input by 2×, and each down-sampling stage in the backbone further downsamples by 2×). The FPN module fuses the two lowest resolution feature maps first by applying a 1x1 conv block + GN to the 16× low-resolution map, up-sampling it by 2× with bilinear interpolation, and adding it to the 8× downsampled feature map. We then perform a similar operation to fuse the 4× downsampled feature map with the newly fused 8× downsampled feature map, and apply a final 3x3 conv to output a feature map of 4× downsampled original resolution with channel dimension 256 as the feature map of per-frame points.

**Attention Block:**   We next provide more details on the feed-forward MLP in each attention block. The feed-forward MLP is consisted of a linear layer with input dimension 256 and output dimension 512, followed by ReLU, DropOut with 10%, a second linear layer with input dimension 512 and output dimension 256 and another DropOut with 10%. We add the output of the MLP to the input of the MLP and return the sum.

**Training Loss Details:** In this section, we provide detailed definitions for our loss functions. The regression loss is defined as:

$$L_{reg}(\{\hat{\mathbf{b}}_i\}, \{\mathbf{b}_i^\star\}) = \frac{\lambda}{M} \sum_i \text{smooth}\ell_1(\hat{x}_i, x_i^\star) + \text{smooth}\ell_1(\hat{y}_i, y_i^\star) + \text{smooth}\ell_1(\hat{l}, l^\star) + \text{smooth}\ell_1(\hat{w}, w^\star)$$

$$+ \frac{1}{M} \sum_i \text{smooth}\ell_1(\sin(2\hat{\theta}_i), \sin(2\theta_i^\star)) + \text{smooth}\ell_1(\cos(2\hat{\theta}_i), \cos(2\theta_i^\star)) \ , \tag{4}$$

with the hyperparameter $\lambda = 0.1$ in practice, and the IoU loss is given by:

$$L_{IoU}(\{\hat{\mathbf{b}}_i\}, \{\mathbf{b}_i^\star\}) = \frac{1}{M} \sum_i \text{IoU}(\text{BBox}(\hat{x}_i, \hat{y}_i, \hat{l}, \hat{w}), \text{BBox}(x_i^\star, y_i^\star, l^\star, w^\star)) \ , \tag{5}$$

to compare the axis-aligned refined and ground-truth bounding box in each frame.

## A.2 Detector and Tracker

To obtain the first-stage coarse initialization, we follow the standard "detect-then-track" approach where a detection model is trained to output per-frame detections and we leverage a tracker to obtain consistent tracklets over time. Next, we give more details about the detector and tracker we use.

**Detector:** To boost detection performance, we adapt the single-frame public implementation of both PointPillars [37] and VoxelNeXt [46] to a multi-frame version that additionally takes 4 past history frames and 4 future frames as input. The validation mean AP of the single-frame vs. multi-frame PointPillars models are 68.78% and 71.02% on the Highway dataset respectively, and 55.98% and 60.58% on AV2 respectively. The validation mean AP of single-frame vs. multi-frame Voxel-NeXt are 81.87%/84.25% on Highway and 60.06%/66.25% on AV2.

**Tracker:** Following [1, 2], we use a simple online tracker, which is largely inspired by [47], and we provide the implementation details of our tracker, in particular how association is performed across frames.

For each new frame at time step $t$ with detections $\mathbf{B}_t = \{\mathbf{b}_t^l\}$ where each $\mathbf{b}_t^l = (x_t^l, y_t^l, l_t^l, w_t^l, \theta_t^l) \in \mathbb{R}^5$ is the individual 2D BEV bounding box, we first filter with Non-Maximum Suppression with IoU threshold 0.1, and then filter out bounding boxes with low confidence scores. We then compute a cost matrix with existing tracklets $\mathbf{S}_t = \{\mathbf{s}_t^j\}$ as follows. For each tracklet $j$, we first predict its bbox position $(x_t^j, y_t^j)$ at time $t$: if the tracklet has at least two past frames, we set $(x_t^j, y_t^j) = 2 * (x_{t-1}^j, y_{t-1}^j) - (x_{t-2}^j, y_{t-2}^j)$ via naive extrapolation (assuming constant velocity between two adjacent frames); otherwise we simply set $(x_t^j, y_t^j) = (x_{t-1}^j, y_{t-1}^j)$. Then, for each pair of the detected bbox $\mathbf{b}_t^l$ and the predicted tracklet bbox $\mathbf{b}_t^j$, we compute the Euclidean distance between the bbox centroids as $\ell^{j,l} = \sqrt{(x_t^j - x_t^l)^2 + (y_t^j - y_t^l)^2}$. For each existing tracklet, we simply employ a greedy strategy to find the nearest detection $l^* = \arg\min_l \ell^{j,l}$, and if the closest distance $l^{j,l^*}$ is greater than a threshold of 5.0m, then the tracklet has no match. We use greedy matching instead of a more sophisticated matching strategy such as Hungarian matching because it is more robust to noisy and spurious detections.

If a tracklet is matched to a new detection, we add the detection to the tracklet and update the tracklet score $c_t^j = \frac{w \cdot c_{t-1}^j + c_t^l}{w + 1.0}$, where $c_{t-1}^j$ is the old tracklet score, $c_t^l = 1.0$ is the detection confidence score we set for every new detection, and $w = \sum_{i=1}^{n_{t-1}^j} 0.9^i$ where $n_{t-1}^j$ is the number of tracking steps in the tracklet.

If a tracklet is not matched, we grow the tracklet by naively extrapolating the position and angle, and set the new confidence score as $c_t^j = 0.9 c_{t-1}^j$.

If a new detection is not matched to any tracklet, we start a new tracklet and initialize the confidence score $c_t^j$ with the detection's confidence.

| Motion State | First-Stage Detector | Second-Stage Refinement | Mean IoU | RC @ 0.5 | RC @ 0.6 | RC @ 0.7 | RC @ 0.8 |
|---|---|---|---|---|---|---|---|
| Stationary | PointPillars | - | 64.46 | 79.34 | 69.95 | 53.58 | 28.66 |
| | | Auto4D | 67.51 | 82.56 | 74.66 | 61.26 | 36.91 |
| | | 3DAL | 68.00 | 81.35 | 75.04 | 63.26 | 41.93 |
| | | Ours (LabelFormer) | **70.67** | **84.08** | **77.31** | **66.03** | **46.81** |
| | VoxelNeXt | - | 68.91 | 83.53 | 75.15 | 61.53 | 38.81 |
| | | Auto4D | 70.87 | 86.11 | 79.06 | 66.28 | 42.86 |
| | | 3DAL | 70.63 | 84.55 | 77.74 | 66.50 | 45.13 |
| | | Ours (LabelFormer) | **73.21** | **86.77** | **80.09** | **69.54** | **51.08** |
| Dynamic | PointPillars | - | 60.53 | 73.12 | 61.04 | 43.23 | 21.03 |
| | | Auto4D | 63.26 | 76.05 | 65.56 | 50.27 | 27.41 |
| | | 3DAL | 60.80 | 72.70 | 60.03 | 45.36 | 22.12 |
| | | Ours (LabelFormer) | **65.64** | **78.06** | **68.69** | **54.74** | **34.10** |
| | VoxelNeXt | - | 62.25 | 73.96 | 62.17 | 46.66 | 25.57 |
| | | Auto4D | 64.73 | 77.01 | 66.77 | 52.02 | 30.72 |
| | | 3DAL | 63.33 | 75.85 | 64.60 | 48.55 | 26.18 |
| | | Ours (LabelFormer) | **66.93** | **79.49** | **69.64** | **56.04** | **35.88** |

Table 4: **[AV2] Performance break-down for ground-truth stationary vs. dynamic objects**

| | Bbox Enc. | Point Enc. | Perturb | Window | # Att | Mean IoU | RC@0.5 | RC@0.6 | RC@0.7 | RC@0.8 |
|---|---|---|---|---|---|---|---|---|---|---|
| $M_1$ | ✓ | | ✓ | All | 3 | 64.20 | 69.83 | 59.62 | 46.11 | 26.58 |
| $M_2$ | | ✓ | ✓ | All | 3 | 69.56 | 82.36 | 75.84 | 64.72 | 44.50 |
| $M_3$ | ✓ | ✓ | | All | 3 | 68.91 | 80.84 | 73.05 | 61.65 | 42.89 |
| $M_4$ | ✓ | ✓ | ✓ | 5 | 3 | 69.07 | 81.19 | 74.17 | 63.47 | 45.05 |
| $M_5$ | ✓ | ✓ | ✓ | 10 | 3 | 69.33 | 81.83 | 74.80 | 64.12 | 45.42 |
| $M_6$ | ✓ | ✓ | ✓ | All | 3 | 70.59 | 83.09 | 75.77 | 65.11 | 46.16 |
| $M_7$ | ✓ | ✓ | ✓ | All | 6 | **70.93** | **83.50** | **76.51** | **66.34** | **47.85** |

Table 5: **[Highway] Ablation study** using the PointPillars initializations. Perturb refers to the bounding box perturbation augmentation, Window specifies the window size when applicable, and # Att is the number of self-attention blocks.

We terminate all tracklets with a tracking confidence score less than 0.1, and apply NMS at the end over all existing tracklets in the current frame with an IoU threshold of 0.1. We repeat this process for the next frame at time $t + 1$ until the end of the sequence.

### A.3 Association with GT Trajectories

For each initial object trajectory detected and tracked in the first stage, we use a simple heuristic to associate it with a ground-truth object trajectory as follows: for each frame that the detected trajectory is present, we identify the ground-truth bounding box that has the maximum IoU with the detected bounding box in that frame. If such ground-truth box has IoU less than 10%, then we fail to find a matching ground-truth box for this frame. As a result we obtain $M'$ ground-truth object IDs for a detected trajectory of length $M$, with $0 \leq M' \leq M$ as we might not be able to find a ground-truth ID for every frame. If $M'$ is 0, then we have failed to find an associated ground-truth object: we consider the detected object as a false positive and discard it in trajectory refinement training and evaluation. Otherwise we take the most common ground-truth actor id out of the $M'$ objects and assign it as the associated ground-truth object trajectory for training and evaluation.

## B  Additional Experiments

**Static vs. Dynamic Objects**    The AV2 validation set contains around 52% stationary objects (we classify an actor as static if the max displacement in the ground-truth displacement in all $X$, $Y$ and $Z$ direction is within 1.0m). Table 4 additionally shows the dynamic vs. stationary object break-down on the AV2 dataset. Our method is able to achieve significantly higher refinement accuracy on both static and dynamic objects with a single network.

**Ablation with PointPillars Init:**    We additionally performed the same set of ablation studies as Table 3 in the main paper on the Highway dataset with the PointPillars-based initializations. Table 5 shows the results, which give the same conclusions as the VoxelNeXt-based initializations in the main paper.

| Architecture | Mean IoU | RC @ 0.5 | RC @ 0.6 | RC @ 0.7 | RC @ 0.8 |
|---|---|---|---|---|---|
| Init | 65.90 | 76.76 | 68.32 | 56.27 | 36.77 |
| MLP (1-layer) | 70.14 | 81.56 | 74.20 | 63.95 | 46.23 |
| MLP (3-layer) | 70.50 | 82.31 | 74.82 | 65.04 | 46.80 |
| MLP (6-layer) | 70.50 | 82.31 | 74.82 | 65.04 | 46.80 |
| LabelFormer(6-block) [44] | **72.38** | **83.68** | **77.45** | **68.33** | **50.06** |

Table 6: **[Highway] Ablation of MLP vs. Transformer** using the VoxelNeXt initializations

| Positional Encoding | Mean IoU | RC @ 0.5 | RC @ 0.6 | RC @ 0.7 | RC @ 0.8 |
|---|---|---|---|---|---|
| Absolute [5] | 71.71 | 83.63 | 76.97 | 67.11 | 48.67 |
| AliBi [44] | **72.38** | **83.68** | **77.45** | **68.33** | **50.06** |

Table 7: **[Highway] Ablation of positional encoding** using the VoxelNeXt initializations

**MLP vs. Transformer** To understand whether attention/transformer-like architecture helps, we ablate the effect of the cross-frame attention module by replacing it with an MLP. Specifically, to aggregate features across frames, we mean pool the per-frame features from all frames, apply an MLP to the pooled features, and then add the aggregated feature back to each frame-level feature. The updated per-frame features are then passed to the decoder module of LabelFormer. For the MLP, each layer consists of a linear layer, followed by LayerNorm and ReLU. We conducted this experiment with 1, 3, 6 layers respectively. The results in Table 6 show that the cross-frame attention module outperforms the MLP architecture by a large margin, demonstrating the benefits of attention with a 40.9% higher relative gain in mean IoU.

**Positional Encoding** We additionally ablate our choice of positional encoding with the VoxelNeXt initialization on the Highway dataset. Table 7 shows that the relative positional encoding AliBi [44] gives overall better performance than using the vanilla absolute positional encodings [5].

## C  Qualitative Results

In this section we show qualitative results for trajectory refinement, comparing LabelFormer with the coarse initialization, 3DAL [2] and Auto4D [1].

We illustrate initial and refined auto-labels for the Highway dataset with VoxelNeXt initializations. Fig. 5 showcases trajectories of two objects on the top that have sparse observations (and hence worse initializations) at the beginning, and denser observations towards the end, and ours LabelFormer is able to give better refinement for the worse initializations because it is able to leverage more temporal context more effectively than previous works. Fig. 5, 6, 7, 8 additionally showcase that our method works better qualitatively on trajectories with both sparse and dense observations and with various speeds. For more visualizations on Argoverse, please refer to the supplementary video.

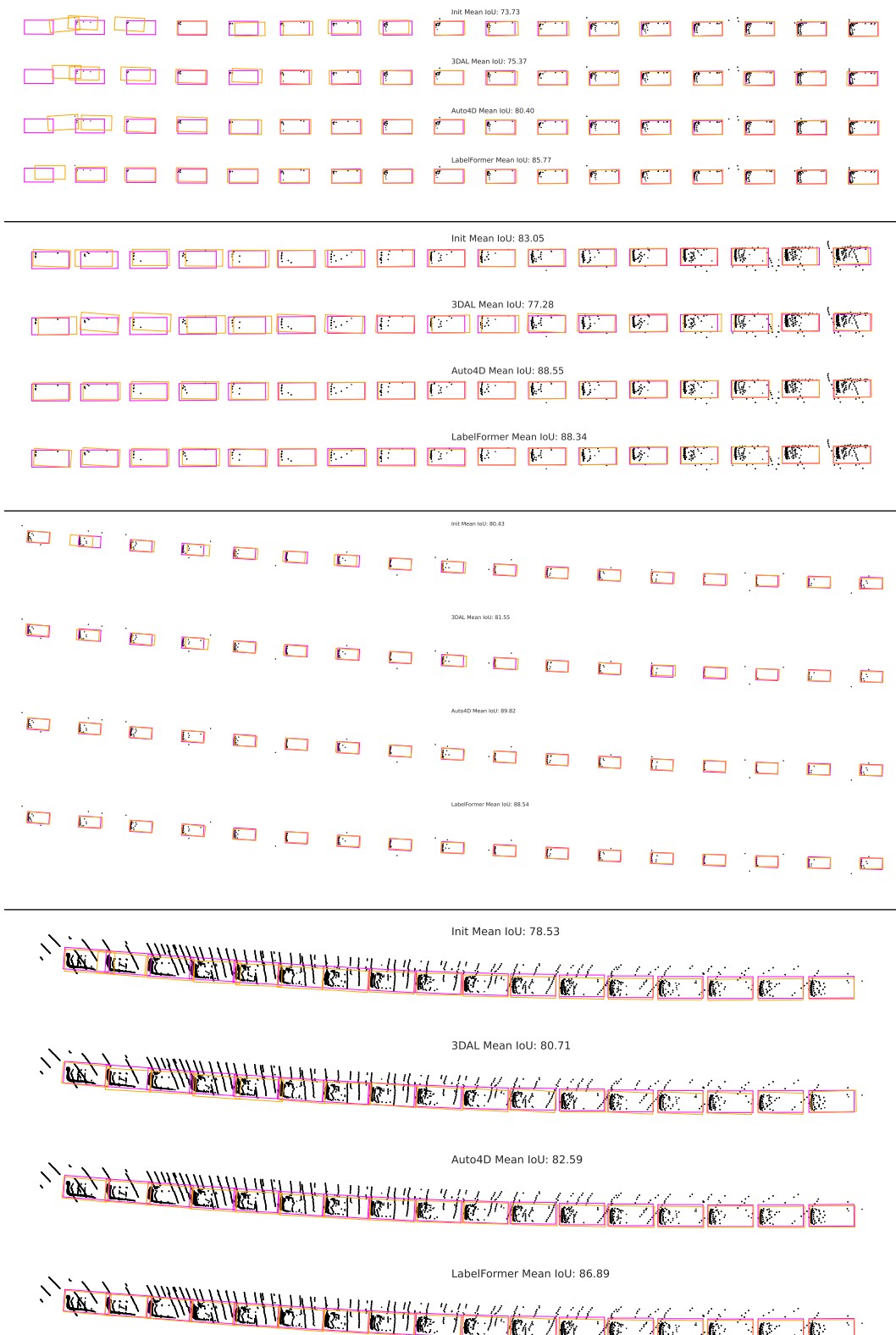

Figure 5: **[Highway] Qualitative results** showcasing different object trajectories (first-stage init, refined by 3DAL, Auto4D and Ours LabelFormer) in each object's trajectory coordinate frame. The ground-truth bounding box is in magenta, and the auto-label is in orange. To avoid cluttering, we visualize every other three bounding box in the first 50 frames of the trajectory.

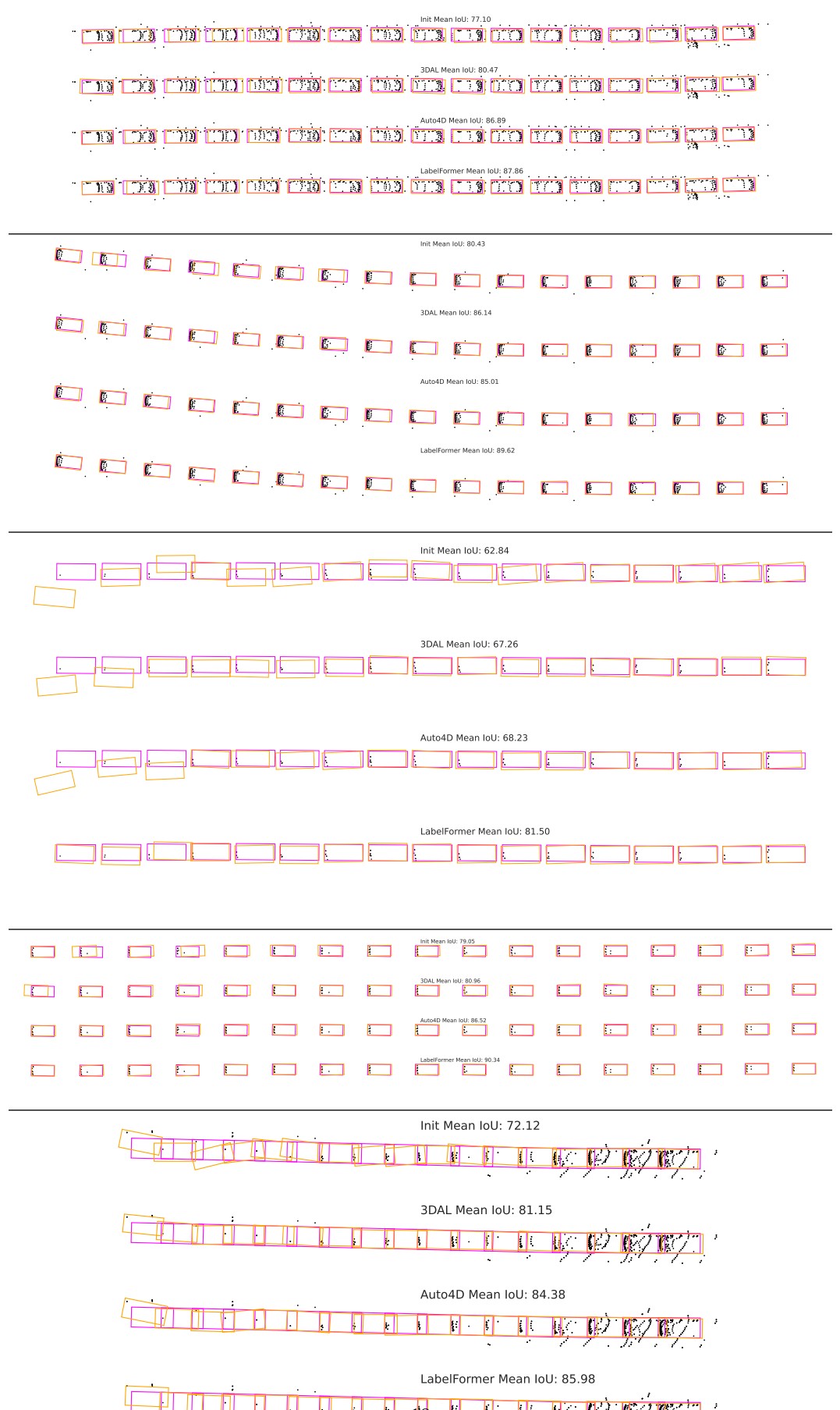

Figure 6: **[Highway] More qualitative results.** Ground-truth in magenta, auto-labels in orange.

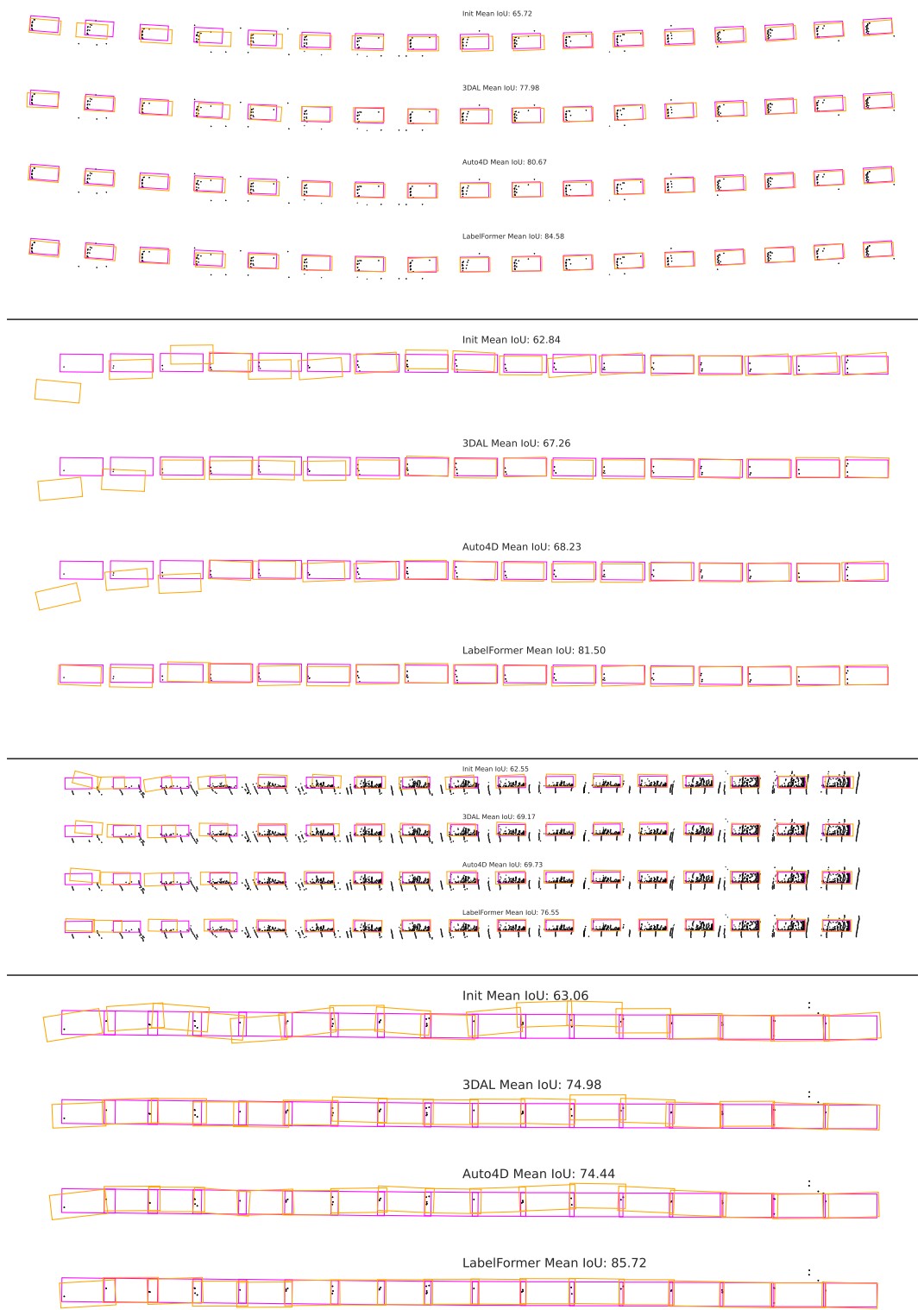

Figure 7: **[Highway] More qualitative results.** Ground-truth in magenta, auto-labels in orange.

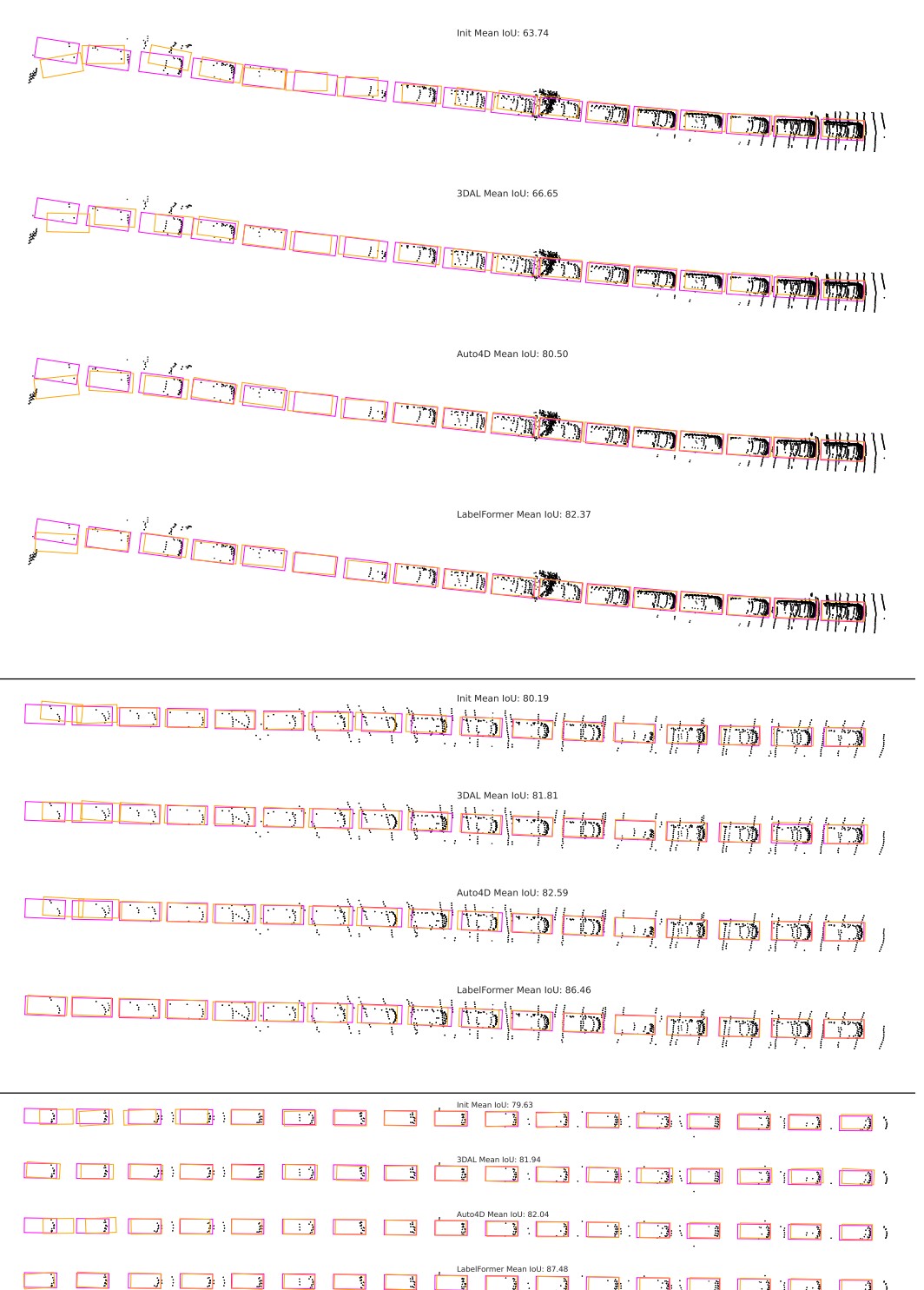

Figure 8: **[Highway] More qualitative results.** Ground-truth in magenta, auto-labels in orange.

