# OpenReview forum: "LabelFormer: Object Trajectory Reﬁnement for Offboard Perception from LiDAR Point Clouds"
_robot-learning.org/CoRL/2023/Conference — CoRL 2023 Poster_

### Official Review · Reviewer_b8wo · 2023-07-17

**Confidence:** 4
**Originality:** Good
**Technical Quality:** Good
**Clarity Of Presentation:** Very Good
**Impact:** 3

**Recommendation:**

Weak Accept: I recommend accepting the paper, but will not argue for my recommendation if the majority of other reviewers have a different opinion.

**Review:**

Strengths:

1. This submission introduces a novel transformer-based architecture for trajectory refinement, which effectively handles both static and dynamic objects, resulting in improved accuracy and efficiency.
2. The paper provides a thorough experimental evaluation on real-world datasets, showing the superiority of LabelFormer over existing methods for trajectory refinement. It also demonstrates the positive impact of using LabelFormer's auto-labels for training downstream object detectors.
3. The paper is generally easy to follow and well organized.

Suggestions:

1. Several related works [44, 45, 46] are missing in the current discussion. Those methods are very related to the proposed methods and therefore should be well compared and discussed.
[44] Pfreundschuh et al., Dynamic object aware lidar slam based on automatic generation of training data, ICRA 2021
[45] Chen et al., Automatic labeling to generate training data for online LiDAR-based moving object segmentation, RAL 2022
[46] Jiang et al., Improving the intra-class long-tail in 3d detection via rare example mining, ECCV 2022

2. As mentioned, this submission mainly focuses on bounding box refinement rather than direct label generation. Furthermore, the proposed transformer-based network still relies on ground truth bounding boxes for supervision. Therefore, I would recommend considering a title that more accurately reflects the focus on trajectory refinement rather than auto-labeling.

3. As outlined in the limitation section, the effectiveness of the proposed method can be significantly affected by the performance of the initial object detection and tracking stage. However, there is currently no clear solution to address the potential consequences of using suboptimal object detection and tracking methods, which may impact the overall results.


**Quality Of The Limitations Section:**

Limitations are not well addressed

**Questions For Rebuttal:**

1. What are the differences between this work from other related studies?
2. Is there any way to enable the proposed trajectory refinement method without relying on ground truth bounding box labels?
3. What are the outcomes when employing suboptimal object detection and tracking methods?

**Robotics Focus:**

Highly relevant to robotics but no hardware experiments

**Summary Of Paper:**

The paper proposes a transformer-based approach called LabelFormer for trajectory refinement to automatically label objects in LiDAR point clouds. The proposed method aims to refine the bounding box trajectory given a noisy initialization obtained from a detect-then-track paradigm. The paper presents multiple experimental results demonstrating the superiority of LabelFormer compared to existing methods in terms of accuracy, efficiency, and downstream detection performance.

**Summary Of Recommendation:**

I would give a weakly accept. The proposed approach is effective and efficient in addressing the trajectory refinement problem. The paper provides a clear presentation of the method and comprehensive evaluation results. However, additional discussion compared to existing methods, and more in-depth ablation studies would further enhance the quality of the paper.

---

### Official Review · Reviewer_pNiA · 2023-07-19

**Confidence:** 4
**Originality:** Fair
**Technical Quality:** Good
**Clarity Of Presentation:** Good
**Impact:** 3

**Recommendation:**

Weak Accept: I recommend accepting the paper, but will not argue for my recommendation if the majority of other reviewers have a different opinion.

**Review:**

The whole problem formulation of automatic labeling has a serious problem or paradox:
  - If this automatic labeling method is so good and can generate nearly perfect object annotations, this problem should have already been considered solved. If that's the case, why not just directly deploy the automatic labeling method in the applications? Why are you still waste your time to annotate data with this method considering the problem has been solved?
  - If this automatic labeling method is not so good and cannot generate nearly perfect annotations, why can this method be used to generate annotations which is supposed to be ground truth and be 100% accurate?

It is true that there have been some previous works published in relevant robotics and/or computer vision venues such as Auto4D that call themselves "automatic labeling", it does not justify that automatic labeling is a valid problem to research. I would argue that methods that study the problem of "automatic labeling" such as Auto4D should never be published.

It would be a valid problem if the authors can call their method "a method for 3D LiDAR point cloud object tracking", "a method for 3D LiDAR point cloud object pose estimation" etc.

---

After rebuttal, the authors addressed the most important concerns of "how the auto annotated is used/evaluated". So I raised my ratings.

**Quality Of The Limitations Section:**

Limitations are addressed clearly

**Questions For Rebuttal:**

The authors can respond to the comments in the review.


**Robotics Focus:**

Highly relevant to robotics but no hardware experiments

**Summary Of Paper:**

The paper introduces a trajectory-level refinement approach for auto-labelling objects in LiDAR point clouds. The approach uses self-attention for reasoning about trajectories, and decodes refined object size and per-frame poses. Evaluation are performed on urban and highway datasets.

**Summary Of Recommendation:**

The studied problem is not a valid problem to study. It is not related to robot learning either.

---

After rebuttal, the authors addressed the most important concerns of "how the auto annotated is used/evaluated". So I raised my ratings.

---

### Official Review · Reviewer_xWt1 · 2023-07-23

**Confidence:** 2
**Originality:** Good
**Technical Quality:** Good
**Clarity Of Presentation:** Very Good
**Impact:** 3

**Recommendation:**

Weak Accept: I recommend accepting the paper, but will not argue for my recommendation if the majority of other reviewers have a different opinion.

**Review:**

The results show noticeable improvements across settings and metrics.  The primary question that's unclear to me is the actual effect of the model choices.

**Quality Of The Limitations Section:**

Limitations are addressed clearly

**Questions For Rebuttal:**

I thank the authors for the inclusion of the ablation studies as the design choices are not immediately obvious from the setting.  That being said, it's not clear the transformer is either as effective as the authors believe or that it's effective for the reasons we want.

1. Given that a single attention layer is nearly equivalent in performance to much more expensive settings (aka 6 layers), do we need attention? Or do we think other factors like the normalization are what actually matters?
2. A similar question can be raised about the effect of the window size.

A simple comparison would be to train a CNN or MLP over a small context window and perform the same training.

Minor: I don't see a discussion of how many heads were used in the attention blocks

**Robotics Focus:**

Sufficient demonstration on hardware

**Summary Of Paper:**

This work presents a trajectory refinement method that is based on detection and tracking of individual objects.  LIDAR automatic labeling across frames allows for augmentation and each sequence is modeled via transformer.  The initial representations are the combination (per frame) of both bounding box and point clouds.  The frames are then decoded (via MLP)  providing a correction.

**Summary Of Recommendation:**

The approach is intuitive and the results convincing but why the model is helpful is unclear and therefore gives me pause about where the innovation really lies.

---

### Decision · Program_Chairs · 2023-08-30

**Decision:**

Accept (Poster)

**Comment:**

The paper proposes a trajectory-level refinement approach for auto-labeling objects in LiDAR point clouds. The reviewers find the paper to be well written and appreciate the thorough experimental evaluations. They raised several different concerns which have been satisfactorily addressed in the rebuttal. Although this paper does not score significantly high in the impact, it is expected to be of interest to researchers in the field.